



# Linking torrential events in the Northern French Alps to regional and local driving atmospheric conditions

Juliette Blanchet[1], Alix Reverdy[1], Antoine Blanc[1,2], Jean-Dominique Creutin[1], Périne Kiennemann[1,3], and Guillaume Evin[4]

[1]Univ. Grenoble Alpes, CNRS, IRD, Grenoble INP, IGE, F-38000 Grenoble, France
[2]Now at RTM-ONF, F-38 000 Grenoble, France
[3]Now at Météo-France, F-74400 Chamonix-Mont-Blanc, France
[4]Univ. Grenoble Alpes, INRAE, CNRS, IRD, Grenoble INP, IGE, 38000 Grenoble, France

**Correspondence:** Juliette Blanchet (juliette.blanchet@univ-grenoble-alpes.fr)

**Abstract.** The Alpine region is strongly affected by torrential floods, sometimes leading to severe negative impacts on society, economy, and the environment. Understanding such natural hazards and their drivers is essential to mitigate related risks. In this article we propose a statistical method based on a mere discriminative index to objectively isolate the atmospheric variables associated with torrential events with long return periods. The method is applied to the Grenoble Metropolitan area in the

Northern French Alps using a list of dates of damaging torrential events since 1851. We consider seven atmospheric variables that describe the nature of the air masses involved and the possible triggers of precipitation, using both 20CRv2c and ERA5 reanalyses. Two spatial scales are considered - a local scale (the Grenoble conurbation) and a regional scale (the French Alps) -, in order to study the spatial variability of the atmospheric signature. This analysis is done conditionally on the main types of generating atmospheric circulation derived from Lamb weather classes, namely the North-West, Southeast-Southwest and

Barometric Swamp classes. The results show that a simple discriminative index - the so-called silhouette index - is able to isolate the variables associated with torrential events, by objectively determining which of them differ particularly from the climatology at the dates of torrential events. All atmospheric variables turn out to be less discriminant for torrential events before 1950 according to 20CRv2c - this is likely linked to 20CRv2c issues over the remote past. For the post-1950 period, in the North-West class - which is both the most frequent class generating torrential events and the best discriminated - humidity

and particularly humidity transport (IVT) plays the greatest role. In the Southeast-Southwest class - the second most frequent class generating torrential events-, instability potential (CAPE) is mostly at play. In the Barometric Swamp class both humidity (PWAT) and instability (CAPE) are discriminant -and even more at the local scale-, showing more mixed situations generating torrential events in this class. The gain in prediction provided by the discriminant variables is substantial. Depending on the class, torrential events are 4 to 14 times more likely when the respective discriminant variables are extreme (typically above

their 0.95-quantile). Although the results are likely to be region-dependent worldwide, the methodology used in this article is generic and could be used elsewhere to find the most discriminating atmospheric variables – provided a list of flooding dates is available. It is also remarkable that the same atmospheric variables with the same discriminative power are found whether we consider them at local or regional scale. This means that, although torrential events are triggered by very local precipitation, the atmospheric signature for such events is actually much wider. Thus, although the present study applies to a small region





of the Northern French Alps, it is reasonable to presume that similar results would apply to other torrential catchments in the French Alps.

# 1  Introduction

The Alps experienced through history numerous disastrous floods (Grazzini, 2007). Orography favors the combination of abundant atmospheric precipitation and fast hydrologic concentration, driven by steep torrents and meandering rivers in flat
glacial valleys. The urban areas situated in valleys are prone to combinations of torrential and riverine floods covering a wide range of vulnerable basin areas (Creutin et al., 2022).

Predicting hazard due to torrential phenomena requires understanding of the driving factors. Many studies showed the link between mesoscale heavy precipitation and atmospheric circulation. Lavers and Villarini (2013) linked mesoscale annual precipitation maxima to the occurrence of atmospheric rivers in Europe (1979-2011). Lenggenhager and Martius (2019) showed
the statistical link between atmospheric blocks in the Euro-Atlantic sector and the frequency of regional-scale heavy precipitation events in Europe. Mastrantonas et al. (2021) showed the prevalence of distinct synoptic-scale atmospheric conditions during the occurrence of heavy precipitation events in the Mediterranean region. At smaller scale, Jacobeit et al. (2009) showed the link between heavy precipitation and circulation types in Central Europe (1850-2003), while Plaut et al. (2001) established strong links between mesoscale heavy precipitation and atmospheric circulation classes in the Alps for the period 1971-1995.
Giannakaki and Martius (2016) identified large-scale flow configurations over Europe that are common to extreme precipitation events in Switzerland. In the Northern French Alps, Garavaglia et al. (2010) and Blanchet et al. (2021b) pointed out the Atlantic and the Mediterranean influences as the main classes of atmospheric circulation associated with extreme precipitation for the period 1950-2020 through respectively zonal and meridional flows. Blanchet et al. (2018), Blanchet and Creutin (2020) and Blanc et al. (2022a) showed that peculiar characteristics of flow strength and flow direction over western Europe drive extreme
precipitation in the Northern French Alps (1950-2017). These studies were all conducted at regional scale or mesoscale, only for precipitation, and mainly for the second half of the twentieth century.

Other studies have directly addressed the link between atmospheric conditions and hydrological extremes. Prudhomme and Genevier (2011) linked circulation type catalogues with flood occurrence for a wide range of catchment sizes (from a few tens of km$^2$ to several thousands) in Europe (1957-2002), in order to determine the most flood-producing circulation types
across regions. Petrow et al. (2009), Bárdossy and Filiz (2005), Jacobeit et al. (2006) and Caspary (1995) applied the same kind of approach to catchment areas from 100 to more than 1,000 km$^2$ (in Western and Central Europe) and for hydrological return times of the order of one year. In particular, Petrow et al. (2009) and Caspary (1995) highlighted a joint change in atmospheric circulations and intense hydrological responses. Froidevaux and Martius (2016) showed through a qualitative analysis of meteorological maps, the importance of water vapor transport in a flow perpendicular to the relief, for regional
scale floods in Switzerland between 1987 and 2011. Still in Switzerland but for a single event, Lenggenhager et al. (2019) studied the blocking activity during the month prior to a high-impact lake flood event that occurred in October 2000. All these studies were interested in meso- to regional-scale watersheds. To the best of our knowledge only Turkington et al. (2014)





studied torrential watersheds of the order of ten square kilometers, by developing atmospheric indicators allowing to isolate the situations generating debris flows and flash floods in the Ubaye region (Southern French Alps). Among all these studies,
only the study by Caspary (1995) goes back further than the middle of the twentieth century, to 1926, but for watersheds of the order of a thousand square kilometers. The study by Turkington et al. (2014) is restricted to the period 1979-2010, during which 64 flash flood events were recorded, corresponding to 6-month return periods. As far as we know, no study applied such an approach at torrential scale before the 1950s, and therefore had a sufficiently large sample of hydrological extremes to study long return periods.

In this article, we take benefit of a database of reported occurrence of damaging torrential flooding going back to 1851 in the Grenoble conurbation (Northern French Alps) (Creutin et al., 2022) to develop a statistical procedure allowing to objectively assess what are the main atmospheric variables driving damaging torrential events with long return periods. We then assess the gain in predicting torrential flooding provided by the most discriminant variables.

Our work makes three contributions to the work of Turkington et al. (2014). First, the studied events are more extreme - the
torrential events correspond to return periods of order 2-3 years at the scale of the conurbation and of 15-170 years at the scale of the torrential units. Second, the driving atmospheric conditions are determined with respect to the main types of atmospheric circulation, which allows us to differentiate the discriminant variables depending of the circulation patterns and thus to improve prediction of torrential events. Last but not least, by taking benefit of two reanalyses with different spatial resolution, we are able to aseess whether the atmospheric signature of torrential events is either local (at the scale of the conurbation) or regional
(at the scale of the French Alps).

## 2 Study region and data

### 2.1 Study region

This article focuses on the Grenoble Metropolitan area, in the French Alps (see the localisation in Figure 2). The torrential watersheds considered intersect the territory of the 75 municipalities included in the Grenoble Metropolitan area and/or the IN-
SEE (National Institute of Statistics and Economic Studies) urban unit of Grenoble. This territory - called the "Y of Grenoble" because of its geometry (Figure 1) - is located at the confluence of two rivers, the Isère and the Drac Rivers, the latter being fed further upstream by a third river, the Romanche River.

In addition to the three river sections separating the limestone mountain massifs of the Vercors and Chartreuse and the crystalline massif of Belledonne, the territory is covered by 139 torrential units as defined by the RTM (Restauration des
Terrains de Montagne, a technical service of the French Forest Administration) (Figure 1). The torrential units show varied geomorphological characteristics with surfaces of less than 1 km$^2$ up to 170 km$^2$ for the Gresse torrent, despite a majority of basins of less than 20 km$^2$. The altitudes are between 180 m and 2977 m.

In the Alps the sources of humidity are multiple, with much water vapor brought from the Mediterranean Sea, although there is a greater contribution from the North Atlantic in winter (Sodemann and Zubler, 2010). From a climatological point of view
the "Y of Grenoble" is at the boundary of two distinct climatic zones: the Northwestern Alps and the Southwestern Alps as





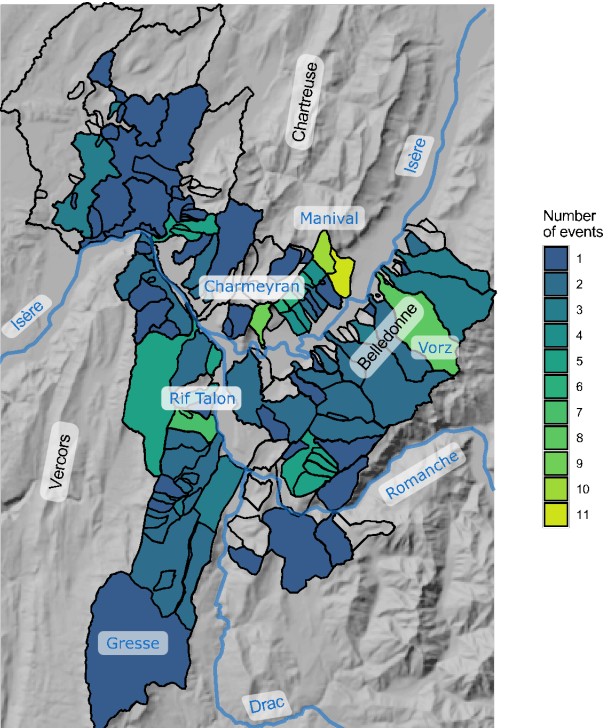

**Figure 1.** Map of the torrential units of the Grenoble conurbation colored according to the number of torrential events observed over the 1851-2019 period.

defined by Auer et al. (2007) using different meteorological variables. With regard to precipitation extremes, this translates into a dominance of Mediterranean flows for the massifs located to the south of our region and oceanic flows for those located to the north of the region (Blanchet et al., 2021b). The "Y of Grenoble" is thus subject to a diversity of atmospheric influences with a strong seasonal variability (Blanchet et al., 2021a, b). This results in a wide range of atmospheric phenomena that can
cause heavy precipitation locally: orographic blocking under a cold front in southerly flow or under a warm front in westerly flow, orographic lifting with dynamic northerly or westerly flows, convective developments by diurnal evolution. This justifies the need of a large sample of torrential events to characterize the generating atmospheric scenarios.

## 2.2 Data

### 2.2.1 Dates of torrential events

The reference for torrential events is given by the database of Creutin et al. (2022). We give below a short summary of its construction but the interested reader is invited to refer to the aforementioned reference for a deeper description. The definition of torrential events is mainly based on the database built by the RTM. The RTM mission of risk mapping in French mountainous areas motivated a systematic reporting for torrential and riverine inundation by trained personal all along RTM existence. The





reports provide quantitative information about the time and place of site events as well as qualitative information about the
generating phenomena and the consequent damages. The reports also graduate semi-quantitatively torrential and riverine events
into respectively 4 (from Very-weak to High) and 3 (from Weak to High) intensity levels. The archive of RTM reports was
made publicly available during the 2010's as data sheet (https://rtm-onf.ign.fr), reporting over 30,000 site events since 1850 all
over the French mountainous areas.

The selection process to define the torrential events at Grenoble Metropolitan scale simply explores chronologically the
RTM database and complements it with historical research data. It looks for concurrence between site events reported by either
the RTM or research data over the "Y of Grenoble" of Figure 1. A set of concurrent site events occurring during neighboring
days is then defined as a Metropolitan hydrometeorological event. The isolated events of category Very-weak are discarded.
The nonisolated events of Very-weak intensities are nevertheless kept - this is open to discussion, but essentially the idea is
to consider that co-occurring floods, even very weak, are "important" events. At the end, the hydrometeorological events of
Creutin et al. (2022) can last from one to several days. In total, the database presents 104 hydrometeorological events between
1851 and 2019.

The events can either involve: a) one or several rivers and no torrent (purely riverine events), b) one or several torrents and no
river (purely torrential events), c) one or several torrent and one or several rivers (mixed events). In total, among the 104 events,
70 belong to b) and c) i.e. involve at least one torrent. These 70 events between 1851 and 2019 (66 events between 1851-2014,
120 42 between 1950-2014, 46 between 1950-2019) are kept for further investigation.  They have return periods of order 2-3 years
at the scale of the conurbation (70 events in 169 years) but of order 15-170 years at the scale of the torrential units (between
1 and 11 events in 169 years, see Figure 1). 20% of these events took place in winter (December and February, DJF), 9% in
spring (March to May, MAM), 59% in summer (June to August, JJA) and 12% in October (September to November, SON). A
majority of the torrential events are multiscale in the sense that they involve multiple torrential units: only 31% of the events
involve only one unit, 21% involve two units, 9% involve three units. Finally, although they can last up to 6 days, the great
majority of them (90%) last no more than 3 days (56% last one day, 24% last 2 days).

In the case of multi-day events, we associate to the event a unique "reference day" defined as the day during which the most
remarkable torrential flood occurred - an expert choice (Creutin et al., 2022). This is both because some nonreference days
correspond to weak events, while our goal is to link the "significant" events to their atmospheric causes, and for statistical
reasons - in order to consider sequences of events of same length. In the case of 1-day events, the reference day is naturally
the event day. Furthermore, to account for the fact that torrential floods can occur early or late in the day -and so be linked to
atmospheric situations the day before or after-, three-day sequences around the reference days are considered (from day-1 to
day+1). We term them the "torrential events" but note that some of them differ in length from the original torrential events of
Creutin et al. (2022). However it is worth noticing that actually very few days of the original torrential events of Creutin et al.
(2022) are discarded with this procedure since only 10% of the original events last more than 3 days.

Finally, each 3-day sequence between 1851 and 2019 is flagged as either an event sequence - when centered around the
reference day of a torrential event -, or a nonevent sequence - otherwise. For example, the 3-day sequence from 1968-12-24 to
1968-12-26 is an "event sequence" if and only if 1968-12-25 (the central day) is the reference day of an event.





### 2.2.2 Weather type classification

In order to characterize the atmospheric conditions driving torrential events, we consider two atmospheric reanalyses which are spatial and temporal interpolations of past meteorological measurements using data assimilation techniques and a meteorological model. 20CRv2c (in short 20CR, Compo et al., 2011) covers the period 1851-2014 with a spatial resolution of 2°. 20CR is composed of 56 individual members that are equiprobable as well as a mean member. In this article, we use the mean member but we also tested the members 1 and 2 (arbitrarily chosen as they are independent) and results turned out to be very similar

(not shown). In addition, in order to study the impact of the spatial resolution, the ERA5 reanalysis (Hersbach et al., 2020), that is available only from 1950 to present, is also studied. This reanalysis has a higher resolution of 0.25° and it assimilates more data, constituting a good reference for comparison. For both databases, daily data are used.

A classification of atmospheric circulation patterns is used to cross with the hydrological response dates of the torrential events of Section 2.2.1. One of the best known and most analyzed method of classifying atmospheric circulation patterns in

synoptic climatology is Lamb Weather Type classification developed for the British Isles by Lamb (1972). Its first development required subjective determination through human assessment of daily weather charts. With the advent of computer science, the classification became more objective.

The Lamb Weather Type objective approach uses gridded daily sea-level pressure (SLP) data and was developed by Jenkinson and Collison (1977). The approach uses three basic variables that define the circulation features at the surface over a given

window: the direction of mean flow, the strength of mean flow and the vorticity. Following previous works by Raynaud et al. (2017), Blanc et al. (2022a) and Blanc et al. (2022b) relating precipitation to surface weather and atmospheric analogues, we consider here the Western Europe window of Figure 2 (see the Lamb points) and, thus, we adapt the method of Jones et al. (1993) to our region of study; we provide details in Appendix A.

The Lamb Weather Type classification contains 27 classes. Since we are here interested in the main types of atmospheric

circulation, we merge the 27 classes into 5 quite balanced classes according to their flow direction and whether they are anticyclonic or not (see Table A2). We obtain the following five classes: North to West (N-W), Southeast to Southwest (SE-SW), East to Northeast (E-NE), High pressure (HP) and Barometric Swamp (BS). The "Barometric Swamp" class corresponds to situations where there is no defined flow and the pressure is relatively uniform spatially and close to average atmospheric pressure. These are days of transition between more marked flow sequences. Figure 2 shows the average surface pressure fields

in each class for ERA5 in 1950-2019. The N-W class is associated with low SLP over Scandinavia and large SLP over the near Atlantic forming a small ridge draining northwestern flows over France. The SE-SW class is associated with a trough over the northeastern Atlantic and quite high SLP over central Europe, draining southern flows over eastern France. High SLP are centered over Great Britain in the E-NE class, while they are found between 30 and 50°N in the HP class, shifting the western flows to the north of France. The BS class is associated with a quite flat SLP field and therefore it corresponds to weak

circulation over Western Europe.

The above classes are obtained at daily scale, for each reanalysis. 3-day sequences are then associated to the majority class if any (86 to 88% of the cases), or to the class of the reference (central) day otherwise.




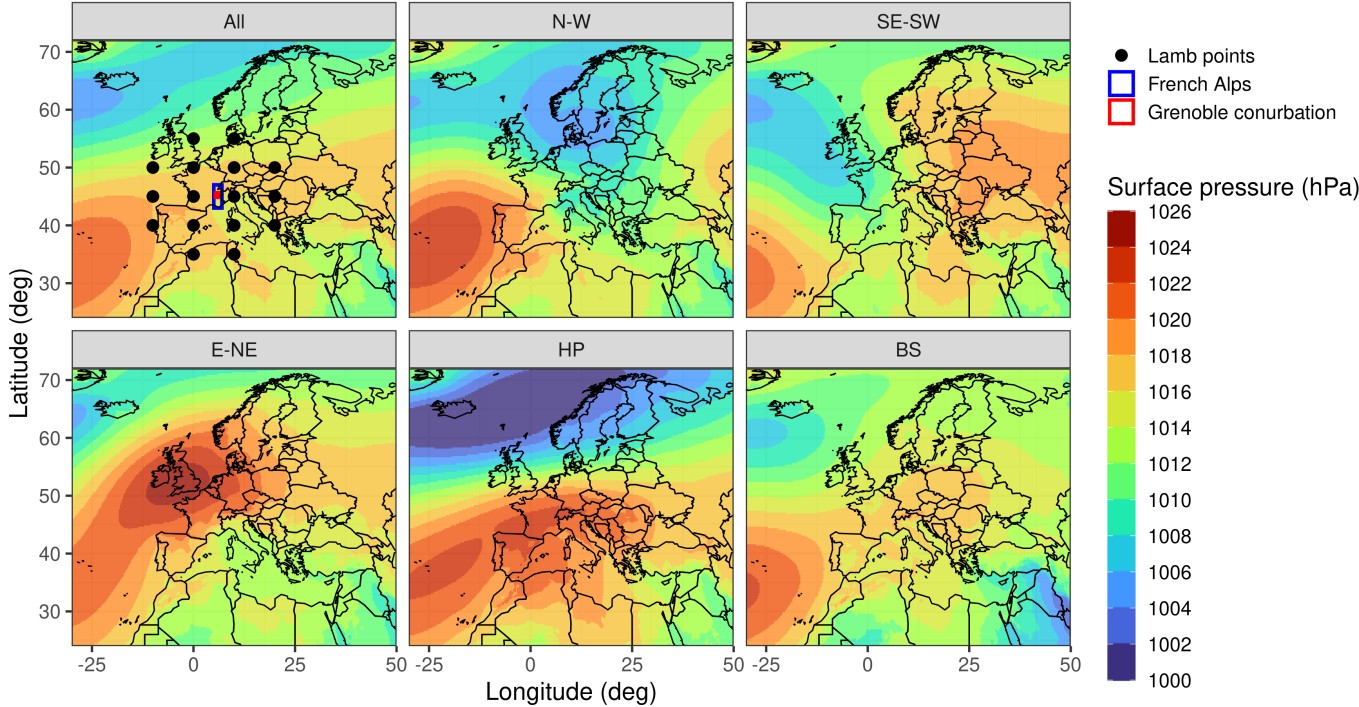

**Figure 2.** Average sea level pressure (hPa) of ERA5 over all the days of 1950-2019 (top left) and the 5 classes derived from Lamb Weather Types. In the "All" case, the black points show the 15 pixels used in Lamb classification, the blue rectangle shows the French Alps and the red one shows the Grenoble conurbation.

### 2.2.3 Atmospheric variables

We consider seven atmospheric variables that describe the nature of the air masses transported and the possible triggers of precipitation: the integrated water vapor column (PWAT), the integrated vapor transport (IVT), the convective available potential energy (CAPE), the specific humidity at 700 hPa ($Q_{700}$), the horizontal wind speed at 700 hPa ($V_{700}$), the pseudo-adiabatic wet bulb potential temperature at 850 hPa ($\theta'_{850}$), and the temperature at 850 hPa ($T_{850}$). The CAPE is a proxy for atmospheric instability that can trigger precipitation (Marsh et al., 2009). Temperature could play a role in the phase of precipitation and melting. $\theta'_{850}$ summarizes the state of an air mass; it can detect anomalously warm and moist air masses (large $\theta'_{850}$) and fronts. Tests have also been conducted on variables related to the relative humidity at 850 hPa and other atmospheric variables at 500 hPa without sufficiently conclusive results to be presented here. PWAT, CAPE, $Q_{700}$ and $T_{850}$ are directly extracted from the reanalyses, while $V_{700}$, IVT and $\theta'_{850}$ are computed following the equations in Appendix B.

In order to summarize the state of the atmosphere and simplify these two-dimensional fields (longitude, latitude), we consider daily spatial averages. Averaging is carried out on two scales in order to study the spatial variability of the atmospheric signature of torrential events: a regional scale - the French Alps (2 pixels for 20CR, 105 pixels for ERA5), and a local scale - the Grenoble





conurbation (9 pixels for ERA5). Note that, as Grenoble is at the border between 2 pixels of 20CR, we cannot consider a smaller region than the whole French Alps for 20CR. In total, four different databases are considered (two reanalyses on different period or window):

- 20CR-1: 20CR in 1851-1949 over the French Alps (2 pixels),

- 20CR-2: 20CR in 1950-2014 over the French Alps (2 pixels),

- ERA5-3: ERA5 in 1950-2019 over the French Alps (105 pixels)

- ERA5-4: ERA5 in 1950-2019 over the Grenoble conurbation (9 pixels).

Comparing the results of 20CR-1 to 20CR-2 allows assessing whether the atmospheric variables driving torrential events have changed over time, according to this reanalysis. We note that the two periods have different lengths (99 versus 65 years),

however considering two equal periods almost does not change the results due to the absence of events in the 1930-1940s. Comparison of 20CR-2 and ERA5-3 allows assessing whether different reanalyses see the same driving atmospheric variables over the recent period and the same region. Note that we have also compared over the common period 1950-2014 and the results are very similar. For shortness and to make the best use of the data, we consider here ERA5 on its full observation period. Finally comparison of ERA5-3 and ERA5-4 allows assessing whether or not the atmospheric signature of torrential

events is stronger at local than regional scales.

## 3   Method

Although applied to a particular region (the Y of Grenoble), the methodology presented below is generic and could be applied elsewhere provided a list of flooding dates. Our goal is to determine which atmospheric variables are very different from the climatology the days of the torrential events. By "very different" we mean rare, compared to random days. Thus, instead

of considering the absolute values of atmospheric variables, we consider their rarity by transforming the variables into non-exceedance probabilities (NEPs) as follows: for each atmospheric variable and for each 3-day sequence, we consider the day that experienced the maximum value. The NEP is the empirical probability of nonexceedance associated to this value with respect to all daily values. A NEP value of $90\%$ for CAPE, for example, means that the maximum CAPE of the sequence is among the 10% largest daily CAPEs. In order to account for potential seasonality, probabilities of nonexceedance are computed

on either the raw daily data or the daily anomalies (substracting the daily climatology from the daily data). Comparison of the results (raw data versus anomalies) will be provided in Section 4. Considering NEPs rather than absolute values can be seen as a way of normalizing the data, since all NEPs are between $0$ and $1$.

A discriminative atmospheric variable is expected to take rare values compared to the climatology during the events and with little variability. Following Turkington et al. (2014), we use the silhouette index, which has been widely used in clustering

techniques to assess the separation between different groups and their tightness (Rousseeuw, 1987). Let consider a given variable and a given atmospheric class, e.g. the raw CAPEs in the N-W class. We divide the corresponding NEPs into two





groups: the 3-day sequences corresponding to a torrential event versus the other 3-day sequences. For a given 3-day sequence, the silhouette index determines how similar its NEP is to other NEPs in its own group compared to NEPs in the other group (Rousseeuw, 1987). For a given 3-day sequence $s$, it is computed as

$$SI(s) = \frac{b_s - a_s}{\max(a_s, b_s)}, \tag{1}$$

where $b_s$ is the average Euclidean distance between the NEP of $s$ and all nonevent NEPs and $a_s$ is the average Euclidean distance between the NEP of $s$ and all event NEPs. Then the silhouette index of torrential events is obtained as the average silhouette index over all torrential events:

$$SI = \frac{1}{N} \sum_{s \in E} SI(s), \tag{2}$$

where $E$ is the set of torrential events and $N$ its cardinal.

Silhouette indices vary between $-1$ and $1$ but we expect them to be near-zero or positive. A silhouette value of 1 indicates that all $a_s = 0$, thus all events have exactly the same atmospheric rarity. A near-zero value indicates that all $b_s \simeq a_s$, thus the NEPs of events and nonevents are quite randomly distributed. In summary, the largest the silhouette, the more similar (grouped) the NEPs of the events. Theoretically, the silhouette does not tell anything on the extremeness of the variable during events - which is yet our goal - since large silhouette could be obtained from nonextreme (but similar) NEPs. However, as will be seen in Section 4, all large silhouettes correspond actually to extreme NEPs so in our application the silhouette does inform on the extremeness.

The silhouette index is computed for each atmospheric class and each variable. The silhouette indices are then compared to each other both within the class to determine which atmospheric variables are the most discriminant for the torrential events of a given class, and in-between the classes to determine which atmospheric class shows the clearest signature. However since the silhouette index is less reliable for unbalanced groups, following Turkington et al. (2014), computation follows a sub-sampling procedure: to make the event and nonevent groups balanced, we calculate the $b_s$ component in Equations 1 and 2 based on a random selection of $N$ (= number of torrential events) nonevent 3-day sequences. We repeat this 1,000 times and keep the average silhouette index over these random draws. This procedure is applied to the atmospheric variables of the four databases.

# 4 Results

## 4.1 The weather types generating torrential events

Before studying the atmospheric variables generating torrential events in the next section, we start with an empirical analysis of the generating weather types. Table 1 shows the distribution of the torrential events across the 5 classes for the different databases. Notably, over the common period 1950-2014, the classification of the torrential events is very concordant across 20CR and ERA5. In particular, the N-W class is very stable - only one N-W event of ERA5 is classified into another class. This is all the more remarkable as over the common period 1950-2014, the overall classification of 20CR differs quite significantly



**Table 1.** Number of torrential events associated to each of the 5 classes. The "20CR & ERA5" case correspond to events that are classified in the same class with both databases.

|  | N-W | SE-SW | E-NE | HP | BS |
|---|---|---|---|---|---|
| 20CR 1851-2014 | 24 | 15 | 5 | 12 | 10 |
| ERA5 1950-2019 | 15 | 15 | 4 | 2 | 10 |
| 20CR 1950-2014 | 13 | 14 | 4 | 4 | 7 |
| ERA5 1950-2014 | 14 | 14 | 4 | 2 | 8 |
| 20CR & ERA5 1950-2014 | 11 | 11 | 4 | 2 | 5 |

**Table 2.** % of 3-day sequences classified in each of the 5 classes.

|  | N-W | SE-SW | E-NE | HP | BS |
|---|---|---|---|---|---|
| 20CR 1851-2014 | 13% | 25% | 18% | 36% | 8% |
| ERA5 1950-2019 | 19% | 25% | 20% | 28% | 8% |
| 20CR 1950-2014 | 12% | 24% | 18% | 37% | 8% |
| ERA5 1950-2014 | 19% | 25% | 20% | 28% | 8% |

from ERA5 for the N-W and HP classes that are mixed up with each other (Table 2). This is not surprising since the HP class corresponds to quite flat pressure fields that are difficult to classify. We can postulate that geopotential fields during torrential events have pronounced enough shapes not to be mixed with HP fields.

Table 1 also shows that very few events are classified in the HP class after 1950 with either database, giving a very low probability of sequences of the HP class to generate torrential events (see Table 3). Furthermore events in the HP class are quite discordant between 20CR and ERA5 - half of the HP events of 20CR over 1950-2014 are otherwise classified in ERA5. For these reasons, the HP class is excluded from the analysis. The E-NE class is also excluded because it contains too few events (see Tables 1 and 3) and it is anyway usually not associated with heavy precipitation events (Blanchet et al., 2021b). As

a result, the rest of this article will focus on the N-W, SE-SW and BS classes, that contain 75 to 87% of the events depending on the reanalysis.

Most of the torrential events are generated by the N-W and SE-SW classes (Table 1). However, since the SE-SW class is the most frequent (Table 2), the most likely classes to generate torrential events are actually the N-W and BS classes (Table 3). Note however that, since we deal with rare events, the weather type classifications have a quite low predictive skill: less

than 0.5% of the sequences of a given class are associated to torrential events. Notably, the great majority of events in the BS class occurs in summer (90% in JJA, see Figure 3). In the other classes, events are more distributed across the seasons, with a majority of events in summer in the SE-SW class (60%) and in winter in the N-W class (50% in DJF).




**Table 3.** Probability (%) that a 3-day sequence of a given class experiences a torrential event.

|  | N-W | SE-SW | E-NE | HP | BS |
|---|---|---|---|---|---|
| 20CR 1851-2014 | 0.31% | 0.10% | 0.05% | 0.06% | 0.20% |
| ERA5 1950-2019 | 0.32% | 0.23% | 0.08% | 0.03% | 0.50% |
| 20CR 1950-2014 | 0.45% | 0.24% | 0.09% | 0.05% | 0.36% |
| ERA5 1950-2014 | 0.32% | 0.23% | 0.08% | 0.03% | 0.43% |

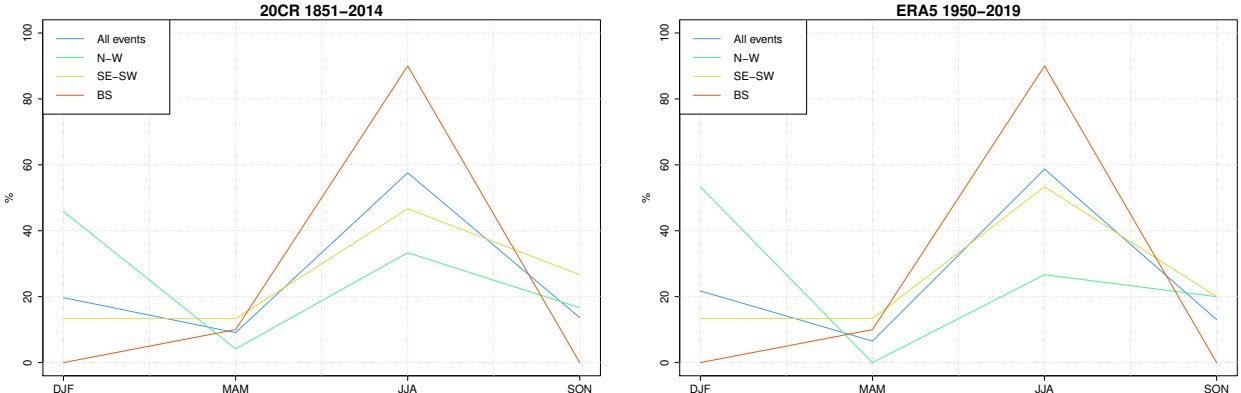

**Figure 3.** Seasonal distribution (%) of torrential events in each class. Left: for 20CR in 1851-2014. Right: for ERA5 in 1950-2019.

## 4.2 The atmospheric conditions driving torrential events

The boxplots of the NEPs during the torrential events are shown in Figure 4 for the different classes, including the all-events
class (irrespective of the classification and including HP and E-NE events) for comparison and as a benchmark since it cor-
responds to the framework of Turkington et al. (2014). A more quantitative comparison of the discriminative power of the
different variables is provided in Figure 5 showing the silhouette indices.

Strikingly, the boxplots of all events in Figure 4 are quite widespread compared to the best variables (i.e. with the largest
NEPs) of each class, which is obviously also visible in the lower silhouette indices of Figure 5. We see here the clear benefit
of considering atmospheric circulation types since different variables seem to discriminate the events depending on the class
- this is a benefit of our work compared to Turkington et al. (2014). Also, differences between the raw and anomaly cases are
relatively slight compared to the between-class differences. This means that the atmospheric variables driving torrential events
are usually rare either overall or for the season.

For the two cases where 20CR over different periods is used (20CR-1 and 20CR-2, for the "All events" and "N-W" cases), we
see a clear difference between the silhouette indices before and after 1950 (blue versus green) with a much better discrimination
(larger silhouette) after 1950. However it is mainly a shift since the best variables are mostly the same over both periods. Note
that the same is observed with two individual members of 20CR (not shown). We can postulate that this is more likely due to
limitations of 20CR in the remote past than a consequence of climate change since there is likely no reason why *all* atmospheric



**Figure 4.** Non-exceedance probability (NEP, %) for each atmospheric variable (from top to bottom: PWAT, $Q_{700}$, $T_{850}$, $\theta'_{850}$, $V_{700}$, IVT, CAPE) depending on both the used database (colors) and the atmospheric classes (panels). Top: for the NEPs of the raw data. Bottom: for the NEPs of the daily anomalies. 20CR-1 is for the French Alps over 1851-1949, 20CR-2 for the French Alps over 1950-2014, ERA5-3 for the French Alps over 1950-2019, ERA5-4 for the Grenoble conurbation over 1950-2019. There is no 20CR-1 boxplot for SE-SW and BS classes due to absence of event before 1950.





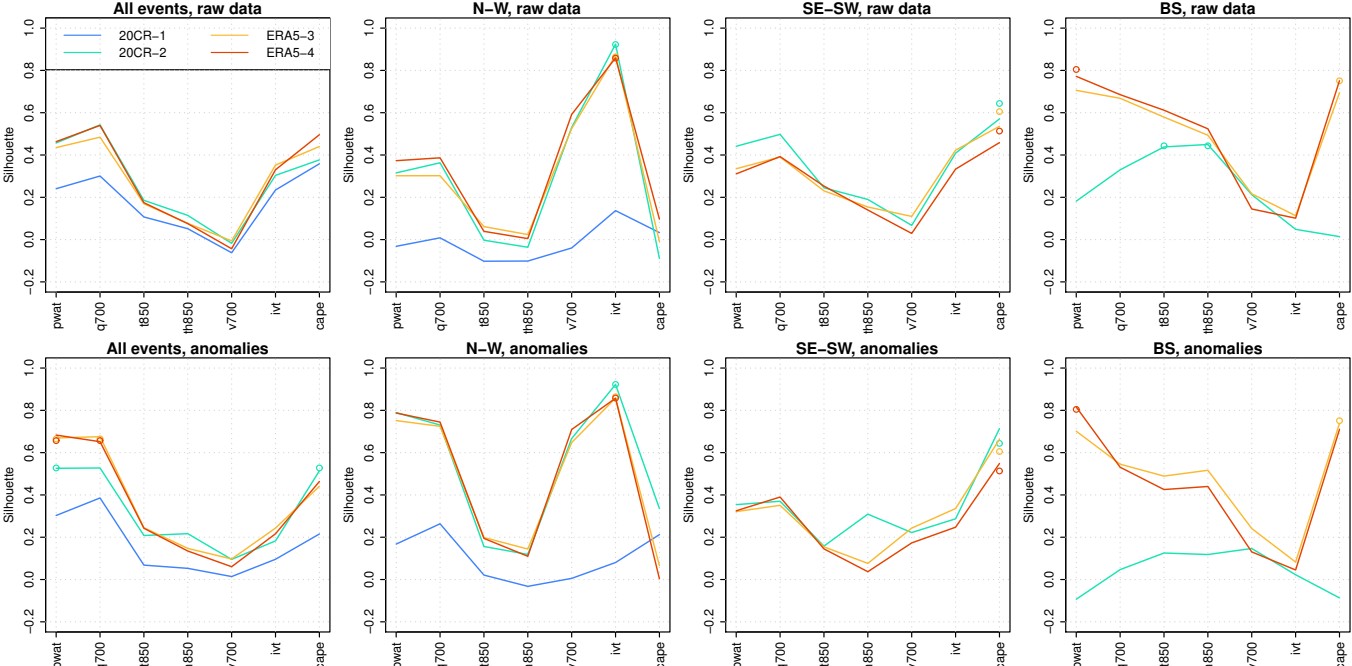

**Figure 5.** Silhouette indices for each atmospheric variable (from left to right: PWAT, $Q_{700}$, $T_{850}$, $\theta'_{850}$, $V_{700}$, IVT, CAPE) depending on both the used database (colors) and the atmospheric classes (panels). Top: for the NEPs of the raw data. Bottom: for the NEPs of the daily anomalies. The points show the largest 2d-silhouettes and the associated pairs of variables (not shown for 20CR-1).

variables should be more discriminant for torrential events after 1950 than before. The reduced amount of assimilated data before 1950 in the 20CR reanalysis (see Fig. 2 of Wang et al. (2013)) may prevent the reanalysis from capturing the rarity of some variables, especially for $V_{700}$ and IVT for the N-W class (Figure 4), leading to lower silhouettes (Figure 5). Given these discrepancies, we were unfortunately unable to study the nonstationarity of the driving atmospheric conditions. Thus, the rest of the paper focuses on the post-1950 period (20CR-2, ERA5-3, ERA5-4).

The silhouette indices of 20CR after 1950 are relatively coherent with ERA5 for all classes but the Barometric Swamp class (green versus yellow), for which 20CR finds no discriminative variable, meaning that all atmospheric variables are looser in 20CR than in ERA5 under Barometric Swamp situations. This is probably partly due to the rougher resolution of 20CR. For all classes, ERA5 gives similar silhouette indices over both the French Alps and the Grenoble conurbation (yellow versus red). This means that, although torrential events are triggered by very local precipitation, the atmospheric signature for such events is actually much wider.

Let now go more precisely into the different classes. Figures 4 and 5 show that, over all events, there is a quite clear signature of both humidity - large values of PWAT and $Q_{700}$, particularly for the season (anomaly) - and instability - large CAPE. Horizontal wind speed ($V_{700}$), temperature ($T_{850}$) and fronts ($\theta_{850}$) are almost not different from the climatology during the events. $Q_{700}$ and CAPE were also found to be the most discriminative variables for torrential events in the Southern French





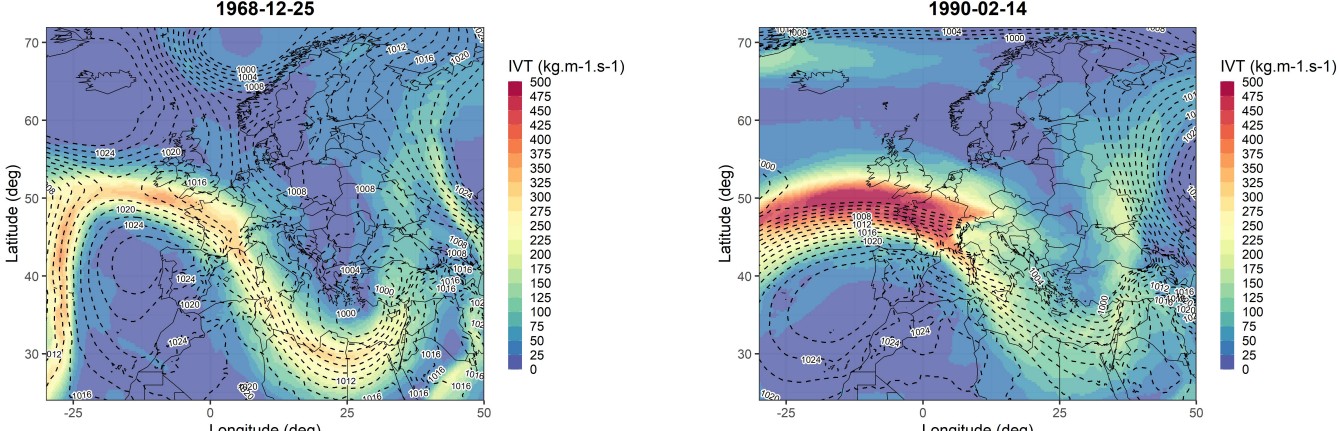

**Figure 6.** IVT fields for 2 events of the N-W class, with ERA5: 1968-12-25 and 1990-02-14. SLP are indicated on the maps and isobars are represented by dotted lines.

Alps in Turkington et al. (2014) (PWAT was not tested), however with lower silhouette indices (around 0.3 against 0.7 here).
One main difference is that here the silhouette is computed over the event class, whereas Turkington et al. (2014) average it over both event- and nonevent-classes. Obviously the NEP of the events in Figure 4 are better grouped (more similar) than that of the nonevents.

The influence of PWAT, $Q_{700}$, IVT and CAPE is more or less pronounced depending on the classes. In the N-W class, the humidity-related variables represented by IVT and the anomalies in PWAT and $Q_{700}$ show a very strong signature (silhouettes
above 0.8) with NEPs mainly above 90%, while the instability represented by CAPE is almost randomly distributed. Notably, the N-W class is the only one showing nonrandomly distributed - and even extreme - horizontal wind speed $V_{700}$ during the torrential events. For illustration, we show in Figure 6 the IVT fields of two events in the N-W class. Only two events are shown but this kind of pattern is found in more than the two third of the N-W events. They are associated with pronounced high pressure systems in the near Atlantic at around 35°N and pronounced low pressure systems in Central Europe. This drives
strong west/northwest flux associated with an elongated and intense water vapor transport towards France. Note that all events of this type in the N-W class occur in winter. The maps of Figure 6 highlight the large scale nature of the atmospheric situations driving torrential events under the N-W class in winter. This type of atmospheric situations have also been highlighted by Blanc et al. (2022a) as driving extreme precipitation over larger catchments of the French Alps (around 5000km$^2$) or by Froidevaux and Martius (2016) for floods in Switzerland, which suggests the relevance of these situations for both medium- and small-scale
catchments.

The SE-SW class also shows rare values of humidity (PWAT, $Q_{700}$, IVT) but to a lesser extent (NEPs mainly above 80%), leading to lower values of silhouette. In this class, a better discrimination is provided by anomalies in CAPE, particularly at the scale of the French Alps (yellow and green), whose NEPs are mainly above 80% during the torrential events. It is difficult to find a recurrent pattern in the fields of CAPE among SE-SW events. We note that all the events associated with large CAPE





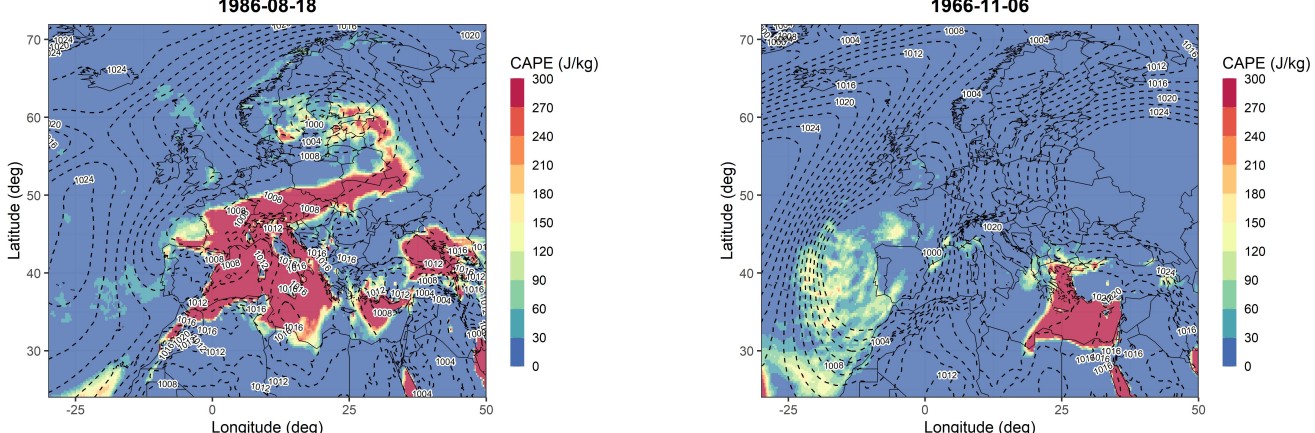

**Figure 7.** CAPE fields for 2 events of the SE-SW class, with ERA5: 1986-08-18 and 1966-11-06. SLP are indicated on the maps and isobars are represented by dotted lines.

values occur from late spring to late summer with a weak low pressure system located over the near Atlantic around the latitude of the Northern Spanish Coast (Figure 7, left). The other events from this class all occur from October to December with more pronounced pressure anomalies and thus larger $V_{700}$ and stronger signature of IVT, but with much smaller CAPE (Figure 7, right). This points to the variety of dynamic/thermodynamic balance through seasons in the SE-SW situations driving torrential events.

The Barometric Swamp class is somehow a mixed case where events show extreme humidity parameters PWAT and $Q_{700}$ but random IVT and very extreme instability (CAPE) according to ERA5. Notably, this is the only class where temperatures $T_{850}$ and $\theta'_{850}$ are not randomly distributed with NEPs mainly above 75%. Events from the BS class do not show recurrent spatial patterns of PWAT or $Q_{700}$, as they are by definition associated with different situations featuring a weak air circulation, as shown for two events in Figure 8. Nevertheless, it is important to note that almost all these events occur in summer (Figure 3), 325  with therefore high air temperature allowing a large water vapor content.

In conclusion the N-W events are characterized by extreme humidity and horizontal wind speed but random instability. This mainly corresponds to intense and elongated water vapor transport from the Atlantic to the Alps in winter. The SE-SW events are characterized by extreme instability but less extreme humidity, mainly occurring in late spring and summer. In comparison, torrential events occurring during Barometric Swamp situations correspond to much more mixed situations with both humidity, 330  instability and temperature being largely "abnormal", occurring systematically in summer. We see here the passage from large-scale atmospheric situations with a clear dynamic signature in cold seasons (N-W in winter and to a lesser extent SE-SW in autumn) to more mixed signatures between dynamic and thermodynamic (SE-SW in summer) and to a diversity of signatures in warm seasons (BS).

The same methodology can be applied to pairs of variables to study concurrent atmospheric hazards. The silhouette indices are still obtained with Equations 1 and 2 but where the distances $b_s$ and $a_s$ are computed over pairs of NEPs. Given the seven



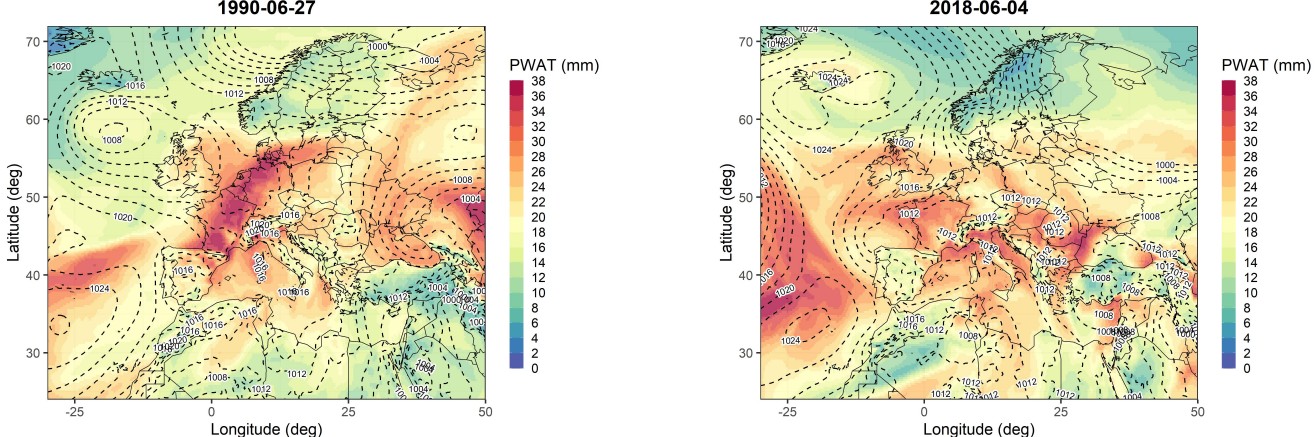

**Figure 8.** PWAT fields for 2 events of the BS class, with ERA5: 1990-06-27 and 2018-06-04. SLP are indicated on the maps and isobars are represented by dotted lines.

variables of study, there are 21 different pairs of variables. Considering the raw and anomaly cases, we obtain, for every day, a total of 42 pairs of NEPs and thus 42 silhouettes. We call them the "2d-silhouettes" to remind that they are associated to pairs, but it must be noted that the values themselves are scalars. Figure 5 points the pairs with largest 2d-silhouettes. There is no surprise here since the best pair of variables is always the two best individual variables and the associated 2d-silhouette

is almost the average of the individual silhouettes. In most cases, the best pairs correspond to a single variable associated to raw data and anomalies (e.g. raw IVT and anomaly IVT in the N-W class). This means that torrential events are mainly linked to a specific variable, which is either large overall (raw) or large for the season (anomaly). The three databases agree on this specific variable in the N-W (IVT) and SE-SW (PWAT) classes.

We now assess the predictive skill of the best pair of variables to predict the occurrence of torrential events. Let $p_1$ and $p_2$ be the pair of NEPs of the two best variables. We compute the empirical probability to experience a torrential event, for a given 3-day sequence featuring NEP $p_1$ larger than a given threshold $u_1$ for the first variable and NEP $p_2$ larger than a given threshold $u_2$ for the second variable:

$$Pr(event|p_1 > u_1, p_2 > u_2).$$

The thresholds $u_1$ and $u_2$ are set to vary from 50% to 99% by step 1%. To assess the gain in prediction, we compare these empirical conditional probabilities to the empirical probability of torrential events in the climatology that is given in Table 3. We compute the ratio, in %, between the two:

$$r(u_1, u_2) = Pr(event|p_1 > u_1, p_2 > u_2)/Pr(event).$$

The corresponding ratios are shown in Figure 9 for the three databases and each class. The images show strong variations

in the top and right borders due to both the low number of events and their over-recurrence for extreme NEPs. The rows/-columns showing constant values of ratios correspond to ranges that do not contain any torrential event but the rightest/topest





pixel. Notably, quite similar ratios are found across databases in the N-W class, whereas the SE-SW class finds larger ratios over the French Alps (20CR-2 and ERA5-3) and the BS class over the Grenoble conurbation (ERA5-4). The largest ratios -corresponding to the largest gains in prediction- are obtained for the N-W class, as expected from the very large silhouettes in

Figure 5. Torrential events are 12 to 14 times more likely when IVT is extreme (above its 0.98-quantile) - whether at regional or local scale and whether in absolute value or anomaly. In the SE-SW class, torrential events are 7 to 8 times more likely when anomaly in CAPE is extreme over the French Alps. This ratio goes down to 5 over the Grenoble conurbation. The BS class shows more variability across databases due to the smaller number of events and the discrepancies between 20CR and ERA5 (Figure 5). Torrential events are about 4 times more likely when PWAT is extreme over the Grenoble conurbation - whether

in absolute value or in anomaly. In summary, depending on the class, torrential events are 4 to 14 times more likely when the respective discriminant variables are extreme - showing the predictive gain of the atmospheric variables compared to a mere weather type classification.

## 5   Conclusions

In this article we have proposed a statistical procedure to objectively determine the atmospheric conditions at the origin of

torrential events with long return periods. This procedure was applied to the Grenoble conurbation using a database of damaging torrential events since 1851. Using two reanalyses of different lengths and different spatial resolutions (20CR and ERA5), our study revealed i) less discriminant atmospheric conditions before 1950 according to 20CR, which is likely due to 20CR limitations in the remote past; ii) a good coherence between the two reanalyses after 1950 with the exception of Barometric Swamp situations. Using a classification derived from Lamb weather types, we were able to show that various atmospheric

conditions are favorable to torrential event occurrence depending on the type of atmospheric circulation. In the N-W class, humidity seems to play the greatest role and in particular IVT which provides substantial gain in prediction compared to the climatology - torrential events are more than 12 times more likely when IVT is extreme. In the SE-SW class, slightly less discriminant atmospheric conditions were found with instability being mostly at play - torrential events are about 7 times more likely when anomaly in CAPE is extreme. Finally for the BS class, stronger discrepancies are found between reanalyses. ERA5

delivers the strongest signature, particularly at local scale, with BS events being characterized by mixed situations where both humidity, instability and temperature are large to extreme.

A remarkable result of our study is that, in the two most frequent classes generating torrential events - the N-W and the SE-SW classes - the discriminant atmospheric variables are as discriminant whether we consider them at local or regional (Alpine) scale. This means that, although torrential events are triggered by very local precipitation, the atmospheric signature

for such events is actually much wider. Thus, although the present study is applied to a small region of the Northern French Alps, we can reasonably presume that similar results would apply elsewhere in the French Alps.

A continuation of this work will be to apply this methodology to climate projections in order to study whether the torrential flood-favorable atmospheric conditions are likely to be more frequent in the future. The fact that we were able to find discriminant signatures at regional scale is a good hope that global climate models such as CMIP6 (Eyring et al., 2016) are enough





**Figure 9.** Ratios $r$ between conditional probabilities of torrential events and climatological probabilities, with respect to the considered thresholds $u_1$ and $u_2$ for the two best atmospheric variables (in abscissa is the best of the two variables according to the univariate silhouettes). Top: 20CR-2. Middle: ERA5-3. Bottom: ERA5-4. The black points show the torrential events. Note that the color scale changes from one class to another.

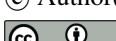



resolved to be used - whether they accurately represent the variables at play is a different story that will obviously have to be checked.

## Appendix A: Weather type classification

The considered weather type classification derives from the Lamb classification. The objective Lamb classification is obtained from thresholds on direction and intensity of the flow and air masses, along their vorticity. This is done using sea level pressure
on 15 points of the reanalysis grid (see 2 according to the equations from Jones et al. (1993). These equations were adapted to account for the different latitudes of our study region in the latitude dependent terms, giving :

$$
\begin{aligned}
F_w &= 0.25(P_{12} + P_{13}) - 0.25(P_4 + P_5) \\
F_s &= 0.25a \times (P_5 + 2P_9 + P_{13} - P_4 - 2P_8 - P_{12}) \\
F &= \sqrt{F_w^2 + F_s^2} \\
D &= \begin{cases} \tan^{-1}(\frac{F_w}{F_s}) \text{ if } F_w < 0 \\ \tan^{-1}(\frac{F_w}{F_s}) + 180 \text{ if } F_w \geq 0 \end{cases} \\
Z_W &= 0.5b \times (P_{15} + P_{16} - P_8 - P_9) - 0.5c \times (P_8 + P_9 - P_1 - P_2) \\
Z_S &= 0.25d \times (P_6 + 2P_{10} + P_{14} - P_5 - 2P_9 - P_{13} - P_4 - 2P_8 - P_{12} + P_3 + 2P_7 + P_{11}) \\
a &= \frac{1}{\cos\phi} \\
b &= \frac{\sin\phi}{\sin(\phi - 5)} \\
c &= \frac{\sin\phi}{\sin(\phi + 5)} \\
d &= \frac{1}{2(\cos\phi)^2} \\
Z_{tot} &= Z_S + Z_W
\end{aligned}
$$

With $P_n$ the surface pressure at point $n$ (in hPa and numbered from West to East and then from North to South), $F_w$ the strength of the Westerly wind component, $F_s$ the strength of its Southerly component and $F$ the total strength of the wind.
$D$ is the direction in degrees of the wind, and $Z_s$ and the $Z_w$ are respectively the Southerly and Westerly components of the shear vorticity, $Z_{tot}$ being the total shear vorticity. $\phi$ is the latitude of the area of interest (45°N in our case for the Grenoble conurbation). Shear vorticity and wind strength are expressed in geostrophic units (hPa per 10° of latitude at 45°N of latitude).

The wind rose is partitioned in 8 different sectors of 45° each, the first being centered on 0°. They are named according to the 8 major directions of wind (N, NE, E, SE, S, SW, W, NW). A direction is attributed according to the value of $D$.
Attribution to weather classes is done according to the rules set in Table A1. The thresholds used to define the no-circulation class (U) are adapted to our latitudes by changing the values from 6 for $Z_{tot}$ and 6 for $F$, to 4.8 for $F$ and 4.2 for $Z_{tot}$. This is





**Table A1.** Criteria of attribution of Lamb weather classes. C means cyclonic, A anticyclonic, N North, E East, S South, W West and U is undefined in the sense that the circulation is weak to non-existent. As a result ANW means Anticyclonic with a North-West component, CC purely cyclonic, etc.

| | $D$ | | | | | | | |
|---|---|---|---|---|---|---|---|---|
| | N | NW | W | SW | S | SE | E | NE |
| $\|Z_{tot}\| < F$ | UN | UNW | UW | USW | US | USE | UE | UNE |
| $F < \|Z_{tot}\| < 2F$ and $Z_{tot} > 0$ | CN | CNW | CW | CSW | CS | CSE | CE | CNE |
| $F < \|Z_{tot}\| < 2F$ and $Z_{tot} < 0$ | AN | ANW | AW | ASW | AS | ASE | AE | ANE |
| $\|Z_{tot}\| > 2F$ and $Z_{tot} > 0$ | CC | | | | | | | |
| $\|Z_{tot}\| > 2F$ and $Z_{tot} < 0$ | AA | | | | | | | |
| $\|Z_{tot}\| < 4.2$ and $\|F\| < 4.8$ | U | | | | | | | |

**Table A2.** The 27 Lamb classes merged into five main classes: North to West (N-W), Southeast to Southwest (SE-SW), East to Northeast (E-NE), High pressure (HP) and Barometric Swamp (BS).

| | |
|---|---|
| N-W: | UN, CN, UNW, CNW, UW, CW |
| SE-SW: | USW, CSW, US, CS, USE, CSE, CC |
| E-NE: | UE, CE, UNE, CNE |
| HP: | AA, AN, ANE, AE, ASE, AS, ASW, AW, ANW |
| BS: | U |

done according to Goodess and Jones (2002) to account for the lesser strength of the wind in our study region compared to the original classification of Jenkinson and Collison (1977) for the British Isles.

Finally the 27 classes are merged into 5 quite balanced classes according to their flow direction and whether they are
anticyclonic or not. Since the no-circulation class (U) does not fit into any other class and is of sufficient size, we keep to alone.. This gives 5 main classes: North to West (N-W), Southeast to Southwest (SE-SW), East to Northeast (E-NE), High pressure (HP) and Barometric Swamp (BS); see Table A2. We call the "U" class "Barometric Swamp" because it corresponds to situations where there is no defined flow and the pressure is relatively uniform spatially and close to average atmospheric pressure.

**Appendix B: Atmospheric variables**

PWAT, CAPE, $Q_{700}$ (specific humidity at 700 hPa) and $T_{850}$ (temperature at 850 hPa) are directly extracted from the reanalyses. The horizontal wind speed at 700 hPa is computed as

$$V_{700} \quad = \quad \sqrt{u_{700}^2 + v_{700}^2},$$





where $u_{700}$ and $v_{700}$ are respectively the zonal and meridional wind components at 700 hPa. The IVT is then computed as

$$IVT = \frac{-1}{g} \times \int Q_P \times V_P \times dP,$$

where $g = 9.81$ m/s$^2$, $P$ is the pressure, $Q_P$ and $V_P$ are the specific humidity and the horizontal wind speed at pressure $P$. Integration is computed on three levels: 850 hPa, 700 hPa et 500 hPa.





$\theta'_{850}$ (in °C) is computed from the specific humidity ($Q_{850}$), the relative humidity ($H_{850}$) and the temperature in K ($T_{850}$) at 850 hPa as follows:

$$\theta'_{850} = \theta_e - C - \exp\left\{\frac{a_0 + a_1 \times X + a_2 \times X^2 + a_3 \times X^3 + a_4 \times X^4}{1 + b_1 \times X + b_2 \times X^2 + b_3 \times X^3 + b_4 \times X^4}\right\} \tag{B1}$$

$$X = \theta_e/C$$

$$C = 273.15$$

$$a_0 = 7.101574$$

$$a_1 = -20.68208$$

$$a_2 = 16.11182$$

$$a_3 = 2.574631$$

$$a_4 = -5.205688$$

$$b_1 = -3.552497$$

$$b_2 = 3.781782$$

$$b_3 = -0.6899655$$

$$b_4 = -0.5929340$$

$$\theta_e = T_{850} \times \left(\frac{1000}{P_d}\right)^{0.285(1-0.28Z)} \exp\left\{Z(1-0.81Z)\left(\frac{3376}{T_{lcl}} - 2.54\right)\right\} \quad \text{(in K)} \tag{B2}$$

$$Z = \frac{Q_{850}}{1 - Q_{850}}$$

$$T_{lcl} = \left\{\frac{1}{T_{850} - 55} - \frac{\ln(H_{850})}{2840}\right\}^{-1} + 55 \tag{B3}$$

$$P_d = P_{lcl} - e_{sw}(T_{lcl})$$

$$e_{sw}(T_{lcl}) = 6.11\exp\left\{19.83 - \frac{5417}{T_{lcl}}\right\} \tag{B4}$$

$$P_{lcl} = 850\left(\frac{T_{lcl}}{T_{850}}\right)^{\frac{C_{pm}}{R_m}}$$

$$R_m = (1 - Q_{850})R_a + Q_{850} \times R_v$$

$$R_a = 287.04 \, J/kg/K$$

$$R_v = 461 \, J/kg/K$$

$$C_{pm} = (1 - Q_{850})C_{pa} + Q_{850} \times C_{pv}$$

$$C_{pa} = 1006.04 \, J/kg/K$$

$$C_{pv} = 1879 \, J/kg/K$$

Equation B1 is from Davies-Jones (2008), Equations B2 and B3 are from Bolton (1980) and Equation B4 is the Auguste-Roche-Magnus formula. $\theta'_{850}$ summarizes the state of an air mass. It is large for a warm and humid air mass.





*Author contributions.* JB: Formal analysis, Funding acquisition, Supervision, Writing – original draft preparation. AR: Data curation, Investigation, Visualization, Writing – review & editing. AB: Investigation, Writing – review & editing. JDC: Conceptualization, Writing – review & editing, PK: Investigation. GE: Funding acquisition, Writing – review & editing.

*Competing interests.* The authors declare there is no competing interests.

455   *Acknowledgements.* This study was partly funded by the European Union via the FEDER-POIA program and thanks to French national funds via the FNADT-CIMA program. This study is also part of a collaboration between the University Grenoble Alpes and Grenoble Alpes Métropole, the metropolitan authority of the Grenoble conurbation (deliberation 12 of the Metropolitan Council of May 27, 2016).





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
