# Peer review of "Linking torrential events in the Northern French Alps to regional and local driving atmospheric conditions"

_Hydrology and Earth System Sciences, 2023_

## Referee Comment (RC2)

**Review – hess-2023-197**

The article isolates discriminant atmospheric variables in combination with differing weather types during torrential events around Grenoble. The findings follow a thorough combination of products and methods and contribute to a better understanding of the atmospheric origin of torrential events. The hydro-meteorological/hydro-climatological approach is part of the scope of the journal. I find that the results are worth a publication in HESS, but I recommend major revisions regarding the placement of the results in the hydro-climatological context and the presentation of the results.

**Major revisions:**

- The article lacks a proper discussion. While parts are interwoven in the results, there is no placement of the results in the hydrological/climatological context. There are similar studies about atmospheric conditions during torrential events in the US or in the rest of Europe. Mentioning and comparing the results with a wider variety of references – other than mainly referring to Turkington et al. (2014) – will provide a valuable overview and add additional strength to the article.
- A clearer structure of the work and some more target-oriented descriptions will help to better convey the results:
    - The results section is very long and interwoven with comparisons and even methods (line 334). This mix blurs the line between results and related work (e.g. line 293/294 appears misleading in the first read) and I would recommend to separate results from discussion and move the method paragraphs to chapter 3.
    - More subchapters could be introduced to the results section. 4.1 and 4.2 might be followed by e.g. atmospheric parameters during specific weather types, local-regional differences or seasonal analyses.
    - Parts of 2.2.2 and 2.2.3 could be considered a part of the introduction, more details are explained in the minor revision section below.
- The claim for results before 1950 should be reduced as the authors themselves conclude, that the data quality is too coarse for real interpretations (e.g. lines 12/13, line 362).

**Minor revisions:**
**Abstract:**

- A more concise abstract would definitely help the reader to grasp the main results quicker.

**1. Introduction:**

- The 2nd and 3rd paragraph are a clean list of work done in the field. It could be rewritten to focus on the content, instead of the authors that looked at it. In other words, the flow of the text is missing (*Author 1 did something. Author 2 did something else…*). Please reformulate to have a more flowing text (e.g. *An interesting finding was this (Author 1), whereas something contrasting was found later (Author 2)…*).
- An introductory paragraph about the general synoptic and atmospheric conditions during torrential events would help to set the scene. What was already known about atmospheric conditions during torrential events? Are they of advective or convective nature?
- The research gap and contributions are clearly formulated ("*our work makes three contributions to the work of Turkington et al. (2014)*"), but should be reformulated with respect

to the general contributions of the study to the scientific field (instead of just referring to Turkington et al. (2014)).

2. Data

- There is a very long description of the torrential events in 2.2.1, about how many days they can last. Based on 2.2.1, I understand, that all major floods are counted as a torrential event. Is this the correct definition? Maybe the events could be grouped: Is there a difference about the origin of the torrential events (caused by advective (often not only local) or convective rainfall, or seasonality)? Are events that might only indirectly be caused by precipitation e.g. snowmelt filtered out of the data set?
- In the abstract it is stated that "*torrential events are triggered by very local precipitatio*n" (line 23/24). Does this match the study´s findings also during west-wind weather patterns and advective rainfall?
- As the seasonality of events is touched e.g. in lines 122/123, 260-262 and 316, did the authors consider a more systematic analysis regarding that? I would expect events in winter being triggered by advective rainfall, westerly weather patterns, and summer events by southern weather patterns and convective rainfall. This could imply e.g higher CAPE values during convective events. It may be worth to clearly analyse the data that way and discuss it in a paragraph or subchapter.
- I understand that the calculation of the pseudo-adiabatic wet bulb potential temperature is rather extensive and moved to the appendix. The horizontal wind speed calculation could, however, be handy to avoid confusion. As it is named $V_{700}$, my first thought was, that it would be based on the v dimension only, forgetting the u dimension. So, I think that it could be clearer to add the formula there, or at least mention that it was calculated from both or change the name to a more general one (e.g. $WS_{700}$ for windspeed at 700 hPa). The text written in the appendix could be good here, also about IVT and $\Theta'850$, but that is up to the preference of the authors.

4. Results

- Maybe colour coding could help Table 1 and 2 to be read more intuitively?
- Table 2 needs a clearer description, that the 3-day sequences are including all moving windows of 3-day sequences, if I understood that correctly (not only the event sequences)?
- Another suggestion would be that Table 1 and 2 could be switched from a logical point of view to move from general to specific to the very specific Table 3.
- Line 251: "*Events in the HP class are quite discordant between the 2 reanalysis products. … For these reasons, the HP class is removed from the analysis.*" This in itself should not be the reason, but that there are only very few events in that group. The reasoning would need some rephrasing.
- In my view, Figure 3 deserves more focus and ideally an entire paragraph or subsection. The seasonal analysis is very hidden, but rather crucial from my understanding.
- The results depicted in Figure 4 are not very clear to me and I struggled to understand their message. So, my suggestion would be: (1) The description of the NEPs should be placed more visible, and maybe it deserves a small reminder while describing Figure 4. Something like "*CAPE values during torrential events lie within the upper half of all values, that generally occur.*" (2) The plot description could be clearer. Are raw data all data and daily anomalies the values during the events? It may be helpful to stick to the same wording throughout the paper.
- The Figures 6-8 are very interesting. Their description could be made clearer and more general, to directly make the point why they are shown. With a clear description, the little conclusion

(line 236-333) should not be necessary anymore. Right now, it helps understanding the point, but the point should be clear from the beginning of the Figure description.

- Figure 9 could use a clearer description of the "best" variables. Maybe the most discriminant? This choice does however limit comparability between the weather type classes and atmospheric parameters. Why is the colour scale not kept the same? Does it not also say something about which parameters function better during some weather type classes than in others?

Appendix A:

- Line 411: "*We keep to alone..*" This probably is an old remainder.

Figures:

- Fig. 1: Maybe something to consider is, that the Figure is difficult to read, when printed in black and white. Please check the Figures following the HESS standards.
- Subplot letters (a, b, c, …) would be handy when referring to the subplots.

---

## Author Comment (AC1)

NB: also not required by the Reviewers, we propose to change the title into "A statistical framework to link the torrential event occurrence to regional and local driving atmospheric conditions – the case of the Northern French Alps", in order to 1) better highlight the fact we conduct a statistical analysis rather than an event-based analysis ("A statistical framework"), 2) better highlight the generality of the method that could be applied elsewhere ("the case of the Northern French Alps"), 3) better stress that we only use dates of events and no other hydrometeorological data ("torrential events occurrence").

**Answer to Reviewer #1**

The current work submitted for review attempts to characterise and identify the critical atmospheric driving conditions of torrential (flash) floods at a local scale using an event dataset collected over the Grenoble Metropolitan area in France and then tries to extrapolate the same over a larger regional scale (Northern French Alps). While the methods applied seem robust enough and the results plausible, I have some significant concerns which should be considered before the paper can be considered for publication in HESS.

My main concern is related to whether the paper is the best fit for the overall scope and target audience of HESS. In my opinion, the current work does not significantly advance the current state of the art in research linked to such extremes and has some critical shortcomings.

=> We thank the Reviewer for his/her thorough review. We regret that the Reviewer does not find that the article significantly advances the current state of research. However, as noted in the introduction, we are aware of only one statistical analysis on the atmospheric conditions generating **torrential** floods - Turkington et al. 2014 - , whereas many works (including those of the co-authors) have been done on the atmospheric conditions generating **riverine** floods. We would be grateful if the reviewer could provide us with further studies on the subject of atmospheric conditions for torrential events, as we indeed claim that the advance of our work is mainly due to the fact that there are gaps on this subject.

Critical Comment

-    In the beginning of Methods (203-204): the authors remark that "*Our goal is to determine which atmospheric variables are very different from the climatology the days of the torrential event*". However, in the 70 events considered for the subsequent analysis (117-118), *b) one or several torrents and no river (purely torrential events), c) one or several torrents and one or several rivers (mixed events)* are considered. Maybe I missed something…wouldn't this lead to a situation where flooding episodes in which riverine processes (saturation excess/snowmelt runoff) played a major role are also included in the torrent event classification? The authors seem to neglect the purely riverine events. In my opinion, the riverine events could have been utilised to overcome the limitations of including the mixed events, or I would have only used purely torrential events.

=> Please let us recall that the considered events are extreme – the torrential events correspond to return periods of order 2-3 years at the scale of the conurbation and of 15-170 years at the scale of the torrential units. We don't think it is possible de get such extreme torrential events due to snowmelt or saturation processes only. Also please note also that mixed events correspond to dates where we had both riverine and torrential events, however the events did not necessarily occur at

the confluence between torrents and rivers (anyway, as shown in Fig 1 of the article, many torrential units have no confluence with rivers). So it seemed reasonable to us to include the mixed events in the analysis because these are dates where something did happen on torrents. Of course, it would also be interesting to understand why some dates are only torrential whereas other are mixed, however we unfortunately don't have for that enough data to conduct a statistical analysis as in this article. Finally, we don't think that including purely riverine events would be helpful because then we would potentially mixed atmospheric conditions triggering flash (torrential) flood over a few hours and atmospheric conditions triggering riverine flood over several days (the time of concentration of the Isère in Grenoble is about 3 days) – these are probably very different conditions. Finally, let us recall that many works have been done on the atmospheric conditions triggering riverine floods, in particular in this region (see e.g. Blanc et al. 2022), whereas works on torrential floods are missing. This is the reason why we wish to focus in this article on floods only.

- The authors state that Comparison of 20CR-2 and ERA5-3 *allows assessing whether different reanalyses see the same driving atmospheric variables over the recent period and the same region*. However, they then proceed to *Furthermore events in the HP class are quite discordant between 20CR and ERA5 - half of the HP events of 20CR over 1950-2014 are otherwise classified in ERA5. For these reasons, the HP class is excluded from the analysis.* Isn't it also worthwhile to check why the coarser scale 20CR and finer ERA5 has difference in the HP class (which from Fig 2 looks to be the only class having a high pressure system over the study area). It is at least expected that the authors provide more explanation on the discrepancy before just ignoring them in subsequent analysis. They also exclude the E-NE class stating it has very few events and is not usually associated with high precipitation; however, that only makes it more worthwhile to explore why such events occur and what are the driving unusual atmospheric conditions behind them.

=> We understand the need for a deeper explanation with HP classes differ between the two reanalyses. However please note that "half of the events" means actually "2 events over 4", so the primary reason why we don't include the HP class is in the previous sentence: the HP class contains too few events to conduct a statistical analysis ("very few events are classified in the HP class after 1950 with either database, giving a very low probability of sequences of the HP class to generate torrential events"). We will investigate the difference for these 2 events but, as already showed in Blanc et al 2022, 20CR fields are smoother and more regular than ERA5 fields. This explains why it classifies too many days in HP, as also shown in Table 2 of the article (37% of 3-day sequences in the HP class in 20CR versus 28% in ERA5). So there is probably nothing more to these two events than the fact that the HP class is overrepresented in 20CR.

We also understand the suggestion of the Reviewer to investigate what are the unusal conditions driving E-NE events, however in this class we only have few events (4-5 events), thus it is again impossible to conduct a statistical analysis as in the rest of the article. The only thing that could be done is to look at the atmospheric variables for the few E-NE events, one by one, and comment on what we see but we fear it would unbalance the article as it would be an "event-based analysis" rather than a "statical analysis" as for the rest of the article. Finally, please note, as stated in the article, that the N-W, SE-SW and BS classes contain 75 to 87% of the events depending on the reanalysis, so by excluding the HP and E-NE classes, we only exclude a minority of the events.

We will clarify these two points in the new version of the article.

- I am also a bit confused about the spatial scales and the averaging used in this study. Are the NEP calculated for each grid point for both 20CR and ERA5 or do you average out all the grids and then calculate the probabilities based on the spatial average time series? Maybe I missed this information; however, this should be emphasised together with the uncertainties involved.

=> We apologize for the confusion. We apply your second alternative: we average out the atmospheric variables over the grids points. This gives us, for every atmospheric variable, a 1-dimensional daily time series. Then we compute the NEP associated to this time-series. We will make it clearer in Section 2.2.3.

- The paper lacks a good discussion section linking the results of the work to the overall research gaps in the field of both extreme events and hydrology. While interdisciplinary studies like this are indeed valuable for the hydrological community, it is expected that the authors will provide a strong discussion linking their atmospheric research to catchment scale and more local water cycle changes.

=> Please note that we have no other data on the torrents than the event dates – this is by the way the reason why we built the database of dates based on the RTM reports, because we have nothing else! There are in the area no series of torrent discharge, only one long-term rainfall station, we don't know precisely the geomorphological characteristics of the torrent catchments, and of course there is no calibrated hydrological model. So unfortunately we are not able to link our results to catchment scales or local water cycle: the only thing we have is a series of dates. This is a very important point that we  did not stress enough in the article. In particular, it'd be impossible for us to conduct an analysis such as that of Prenner et al. 2019 that relies on many catchment data. Our analysis can be seen as a "low-cost" analysis where only dates are necessary to link them with atmospheric conditions. We will clarify this important point in the Data section of the article, as well as in the last part of the introduction that will be partly rewritten.  We also propose to mention "torrential event occurrence" in the title to make clearer that we use dates only.

However, we understand the point of view of the Reviewer that the paper lacks a clear discussion. Actually, in the current version the results and discussion are mixed in Section 4, which we understand is confusing. Thus we propose to split Section 4 in 2 parts: a "Results" Section that would only be about the NEP values (including Figs 4, 5, 9), and a "Discussion" Section that would comment on the maps of the discriminant variables of each class (including Figs 6, 7, 8) and discuss how they differ from what we know of riverine flooding in the literature. We hope this will clarify the article.

I also had a feeling that some of the content of the manuscript seems very clear and straightforward to the authors but is not so clear to more general readers. For example, Figure 1 doesn't follow the acceptable standards for publication in an international journal. The north arrow and scale bar are missing.  It is not clear to me what a torrential unit means. Is it linked to the geomorphology of the study region, or is it linked to the actual event reporting convention followed by the agency?

=> You're right, we will add in Figure 1 the north arrow and a scale bar and go through the article to see if any other information is missing. The torrential units are a definition of the agency.

- In the conclusions, the authors noted that – "*A remarkable result of our study….. the discriminant atmospheric variables are as discriminant whether we consider them at local or regional*". Are these results really remarkable? The authors themselves note that (185-190), as

Grenoble is at the border between 2 pixels of 20CR, we cannot consider a smaller region than the whole French Alps for 20CR. Doesn't this mean we move ahead with the assumption that local-scale site specific atmospheric conditions are indeed related to the dynamics captured by the coarser scale product? I would recommend defining the local and regional scale more strictly here to close these inconsistencies. Furthermore, spatially averaging (which is not well defined in this paper) at the so-called local and regional scales would add significant uncertainties to this methodology. By default, the spatial averages at both scales should indeed contain information complimentary to each other.

=> Actually the "remarkable result" relates to the comparison of ERA5-Alps and ERA5-Grenoble. In the first case, we average the ERA5 atmospheric variables over the 105 pixels of the French Alps before computing the NEPs. In the latter case, we average the ERA5 atmospheric variables over 9 pixels covering Grenoble conurbation before computing the NEP values. Remarkably, in both cases, we find the same discriminant atmospheric variables and with very similar Silhouette values (see Fig 5, yellow vs. red curves). This means that, although we consider date of very local events (the torrential units scale a few km²), we find the same "abnormal" atmospheric variables whether we look at them at the scale of the French Alps or at the scale of Grenoble. This seemed remarkable to us and actually we did not expect that (we expected less discriminant NEP values at the French Alps scale). If the word "remarkable" causes concern, we can replace it with "unexpected".

- The authors end the Introduction by stating – "*Our work makes three contributions to the work of Turkington et al. (2014).*" I expected a much more concise presentation about the work and its proposed relevant outcomes for understanding hydrological functioning rather than simply stating how this work is different from another similar study carried out around nine years ago. Furthermore, the first advancement is confusing *(As far as we know, no study applied such an approach at torrential scale before the 1950s*), as it is later stated that the period before 1950 is not being investigated fully due to inconsistencies in the 20CR model (*Given these discrepancies, we were unfortunately unable to study the nonstationarity of the driving atmospheric conditions. Thus, the rest of the paper focuses on the post-1950 period*)

=> We understand your point of view and this part (last paragraph of the Introduction) will be deeply rewritten.

Minor comments:

- In conclusion, the authors again stater *The fact that we were able to find discriminant signatures at regional scale is a good hope that global climate models such as CMIP6.* To my understanding, the grid resolution of CMIP6 is at least comparable to the resolution of the 20CR model used here. Why not use the historical CMIP6 model ensemble directly at the so-called regional scale?

=> Actually historical CMIP6 simulation do not reproduce the true climatology (they are simulations) so we could not link them to actual dates of events. Anyhow, ERA5 is of better resolution and it's the only database that allows to compare two spatial scales (French Alps vs Grenoble).

- The abstract needs to be restructured for better readability. I would omit the part about applying the method elsewhere..

=> We will work on the abstract to make it more readable and more concise.

- The terms ERA5-3 and ERA5-4 is actually confusing because you also talk about 3 day windows.. I would recommend naming them as ERA5-A for Alps and ERA5-G for Grenoble.

=> Thanks, this is a good suggestion.

- *We note that the two periods have different lengths (99 versus 65 years), however considering two equal periods almost does not change the results due to the absence of events in the 1930-1940s* – It is not clear to me what is meant here.

=> We meant here that, since we're only interested in the NEP values of the torrential dates, considering 20CR on 1851-1949 (99 yrs) vs 1950-2014 (65 yrs), or on 1851-1932 (82 yrs) vs 1933-2014 (82 yrs) would barely change the number of points in e.g. Fig 4 since there is almost the same number of events in 1851-1949 and 1851-1932. But anyway, in order to simplify the article and to make it less confusing, we will remove the analysis of 20CR prior to 1950 since anyway we conclude that the quality of 20CR in the remote past is less good, so this part will disappear. Thus we will in the next version focus on three databases: 20CR-Alps on 1950-2014, ERA5-Alps on 1950-2019, ERA5-Grenoble on 1950-2019. Another possibility to make the article shorter could actually be to keep ERA5 only (ERA5-Alps on 1950-2019, ERA5-Grenoble on 1950-2019) since anyway 20CR gives the same results as ERA5-Alps.

- In Figures 6-8, the authors only show one atmospheric variable (IVT, CAPE & PWAT) for each class (NW, SE-SW and BS respectively), however, it would be interesting to see the three variables for all the three events so that a ready comparison is possible. This could help strengthen the argument about different drivers for different events.

=> We understand you're point of view. We will show in the next version other variables than the discriminant ones but note that we may have to make a selection since there are 7 variables in total and showing all will make big figures.

- I could also find quite some missing words and grammatical inconsistencies. The draft could benefit from a fresh reading to smoothen out such errors.

=> We will carefully go through the article to check for typos and errors.

Overall, in view of the critical shortcomings, I recommend the paper to be rejected.

=> We sincerely respect your recommendation, and we thank you again for your comprehensive review. However it seems to us that the above comments could be addressed in the next version of the article. I hope you will find our replies convincing.

References
Blanc, A.; Blanchet, J. & Creutin, J.-D.
Characterizing large-scale circulations driving extreme precipitation in the Northern French Alps
*International Journal of Climatology,* **2022***, 42,* 465-480

Prenner, D.; Hrachowitz, M. & Kaitna, R.
Trigger characteristics of torrential flows from high to low alpine regions in Austria
*Science of The Total Environment,* **2019***, 658,* 958-972

---

## Author Comment (AC2)

NB: also not required by the Reviewers, we propose to change the title into "A statistical framework to link the torrential event occurrence to regional and local driving atmospheric conditions – the case of the Northern French Alps", in order to 1) better highlight the fact we conduct a statistical analysis rather than an event-based analysis ("A statistical framework"), 2) better highlight the generality of the method that could be applied elsewhere ("the case of the Northern French Alps"), 3) better stress that we only use dates of events and no other hydrometeorological data ("torrential events occurrence").

**Answer to Reviewer #2**

The article isolates discriminant atmospheric variables in combination with differing weather types during torrential events around Grenoble. The findings follow a thorough combination of products and methods and contribute to a better understanding of the atmospheric origin of torrential events. The hydro-meteorological/hydro-climatological approach is part of the scope of the journal. I find that the results are worth a publication in HESS, but I recommend major revisions regarding the placement of the results in the hydro-climatological context and the presentation of the results.

=> We thank the Reviewer for his/her encouraging review.

Major revisions:
- The article lacks a proper discussion. While parts are interwoven in the results, there is no placement of the results in the hydrological/climatological context. There are similar studies about atmospheric conditions during torrential events in the US or in the rest of Europe. Mentioning and comparing the results with a wider variety of references – other than mainly referring to Turkington et al. (2014) – will provide a valuable overview and add additional strength to the article.

=> We understand the point of view of the Reviewer that the paper lacks a proper discussion and this was also noted by Reviewer #1. As stated, in the current version the results and discussion are mixed in Section 4, which we understand is confusing. Thus we propose to split Section 4 in 2 parts: a "Results" Section that would only be about the NEP values (including Figs 4, 5, 9), and a "Discussion" Section that would comment on the maps of the discriminant variables of each class (including Figs 6, 7, 8) and discuss how they differ from what we know of flooding in the literature. However, although we are aware of many works on the atmospheric conditions generating riverine flooding (including those of the co-authors of this article), we aren't aware of much work on the subject for torrential floods in Europe. There is of course Turkington et al. (2014) in France that we cite a couple if times and to which we could compare. There is also Prenner et al. 2019 in Austria but their work differ from ours since they link torrential flows to hydrometeorological processes (long-lasting rainfall, short-duration storm, intense snow melt) rather than to atmospheric conditions (please note that we don't have the hydrometeorological data to conduct such an analysis in our region). There is a more active literature on torrential rainfall to which we could probably compare although torrential rainfall and torrential floods can differ. In particular, we have found a quite comprehensive literature on torrential rainfall in Spain (e.g. Milan et al. 1995 – but linking torrential rainfall to sea surface temperature that we do not consider in this article; Grimalt-Gelabert et al. 2021 – but linking torrential rainfall to weather types as the LWT of this article, rather than to atmospheric variables). There is also a very comprehensive literature on heavy rainfall in the Mediterranean Basin, e.g. the review of Dayan et al. 2015 that links heavy rainfall to atmospheric processes at different scales. We will make our best to compare our results to these studies but to the best of our knowledge there is a hole in knowledge regarding torrential floods. We would be grateful if the reviewer could provide us with further studies on the subject he/she is aware of.

- A clearer structure of the work and some more target-oriented descriptions will help to better convey the results:

o The results section is very long and interwoven with comparisons and even methods (line 334). This mix blurs the line between results and related work (e.g. line 293/294 appears misleading in the first read) and I would recommend to separate results from discussion and move the method paragraphs to chapter 3.

=> As said above, we agree with this suggestion and we will separate results and discussion in separate sections.

o More subchapters could be introduced to the results section. 4.1 and 4.2 might be followed by e.g. atmospheric parameters during specific weather types, local-regional differences or seasonal analyses.

=> Well noted, we will reorganize the results section.

o Parts of 2.2.2 and 2.2.3 could be considered a part of the introduction, more details are explained in the minor revision section below.

- The claim for results before 1950 should be reduced as the authors themselves conclude, that the data quality is too coarse for real interpretations (e.g. lines 12/13, line 362).

=> You're right. We will remove the analysis of 20CR prior to 1950 since anyway we conclude that the quality of 20CR in the remote past is less good. Thus we will in the next version focus on three databases: 20CR-Alps on 1950-2014, ERA5-Alps on 1950-2029, ERA5-Grenoble on 1950-2019. The mentioned lines will thus be removed.

Minor revisions:
Abstract:
- A more concise abstract would definitely help the reader to grasp the main results quicker.

=> We will work on the abstract to make it more concise.

1. Introduction:
- The 2nd and 3rd paragraph are a clean list of work done in the field. It could be rewritten to focus on the content, instead of the authors that looked at it. In other words, the flow of the text is missing (Author 1 did something. Author 2 did something else…). Please reformulate to have a more flowing text (e.g. An interesting finding was this (Author 1), whereas something contrasting was found later (Author 2)…).

=> We agree with this suggestion and we will reformulate these parts.

- An introductory paragraph about the general synoptic and atmospheric conditions during torrential events would help to set the scene. What was already known about atmospheric conditions during torrential events? Are they of advective or convective nature?

=> Torrential events in the Alps are mainly convective in summer and spring (this concerns most of the events). In autumn, events can be either convective or advective, while in winter they are mainly advective. We will add a paragraph on what is known about atmospheric conditions during torrential events but, as already mentioned, little seems actually to be known for the Alps. We will however state what is known about torrential rainfall since there is a wider literature on this subject (see the aforementioned references).

- The research gap and contributions are clearly formulated ("our work makes three contributions to the work of Turkington et al. (2014)"), but should be reformulated with respect to the general contributions of the study to the scientific field (instead of just referring to Turkington et al. (2014)).

=> We agree with that and this was also mentioned by the Reviewer #1. We will reformulate this part.

2. Data
- There is a very long description of the torrential events in 2.2.1, about how many days they can last. Based on 2.2.1, I understand, that all major floods are counted as a torrential event. Is this the correct definition? Maybe the events could be grouped: Is there a difference about the origin of the torrential events (caused by advective (often not only local) or convective rainfall, or seasonality)? Are events that might only indirectly be caused by precipitation e.g. snowmelt filtered out of the data set?
=> First, all "major floods" are considered as events but only those involving torrents are considered in this study (purely riverine events are discarded because they involve different spatial and temporal scales as the torrential events). Second, we do not filter the events that could be caused by snowmelt but 1) let us recall that the considered events are extreme – the torrential events correspond to return periods of order 2-3 years at the scale of the conurbation and of 15-170 years at the scale of the torrential units. We don't think it is possible de get such extreme torrential events due to snowmelt or saturation processes only. Extreme rainfall is a prerequisite and snowmelt is only the cherry on the top (see Marra et al. 2016). 2) We actually don't have the information about what caused the event. Actually, we have no data else than the date of events. Our analysis merely links torrent event occurrence to the atmospheric conditions, without any other data. This is an important point that we will clarify in the Data section of the article, as well as in the last part of the introduction that will be partly rewritten. We also propose to mention "torrential event occurrence" in the title to make clearer that we use dates only. For this reason, we are not able to group the events according the type of rainfall. Finally, grouping the events by season would lead to too small groups apart for N-W in winter and SE-SW in summer, so only these two cases could be studied. And even there, subsampling by weather types would lead to too small groups. This is probably debatable but we prefer keeping the weather pattern subsampling because it gives interesting results regarding the different atmospheric variables at play depending on the weather type.

- In the abstract it is stated that "torrential events are triggered by very local precipitation" (line 23/24). Does this match the study´s findings also during west-wind weather patterns and advective rainfall?
=> Summer and spring torrential events are mainly convective, so they are triggered by local precipitation. In winter, they are mainly advective corresponding to widespread rainfall but that can be larger in some parts. We will repharse this sentence to make it clearer.

- As the seasonality of events is touched e.g. in lines 122/123, 260-262 and 316, did the authors consider a more systematic analysis regarding that? I would expect events in winter being triggered by advective rainfall, westerly weather patterns, and summer events by southern weather patterns and convective rainfall. This could imply e.g higher CAPE values during convective events. It may be worth to clearly analyse the data that way and discuss it in a paragraph or subchapter.
=> You're right, summer events are mainly convective with southern WPs and winter events mainly advective with westerly WPs. We will complement Figure 3 with a figure showing the % of events classified in each class for each season to make that more explicit. However, as mentioned above, grouping the events by season would lead to too small groups apart for N-W in winter and SE-SW in summer, so only these two cases could be studied. This is probably debatable but we prefer keeping the weather pattern subsampling because it gives interesting results regarding the different atmospheric variables at play depending on the weather type.

- I understand that the calculation of the pseudo-adiabatic wet bulb potential temperature is rather extensive and moved to the appendix. The horizontal wind speed calculation could, however, be handy to avoid confusion. As it is named V700, my first thought was, that it would be based on the v dimension only, forgetting the u dimension. So, I think that it could be clearer

to add the formula there, or at least mention that it was calculated from both or change the name to a more general one (e.g. WS700 for windspeed at 700 hPa). The text written in the appendix could be good here, also about IVT and Θ'850, but that is up to the preference of the authors.
=> We will change the notation of V700 to WS700 to avoid confusion.

4. Results
- Maybe colour coding could help Table 1 and 2 to be read more intuitively?
=> Please note that Tables 1 to 3 will be shorter in the next version because we will remove the 20CR case on 1851-2014.

- Table 2 needs a clearer description, that the 3-day sequences are including all moving windows of 3-day sequences, if I understood that correctly (not only the event sequences)?
=> That's true, Table 2 shows all moving window 3-day sequences. We will clarify this.

- Another suggestion would be that Table 1 and 2 could be switched from a logical point of view to move from general to specific to the very specific Table 3.
=> This is a good suggestion, we will start by Table 2, then Table 1 and finally Table 3.

- Line 251: "Events in the HP class are quite discordant between the 2 reanalysis products. … For these reasons, the HP class is removed from the analysis." This in itself should not be the reason, but that there are only very few events in that group. The reasoning would need some rephrasing.
=> We agree with that, the true reason is that it contains too few events. We will reformulate this part.

- In my view, Figure 3 deserves more focus and ideally an entire paragraph or subsection. The seasonal analysis is very hidden, but rather crucial from my understanding.
=> We understand your point of view that the seasonality is important. However, as mentioned above, grouping the events by season would lead to too small groups apart for N-W in winter and SE-SW in summer, so only these two cases could be studied. This is probably debatable but we prefer keeping the weather pattern subsampling because it gives interesting results regarding the different atmospheric variables at play depending on the weather type.

- The results depicted in Figure 4 are not very clear to me and I struggled to understand their message. So, my suggestion would be: (1) The description of the NEPs should be placed more visible, and maybe it deserves a small reminder while describing Figure 4. Something like "CAPE values during torrential events lie within the upper half of all values, that generally occur." (2) The plot description could be clearer. Are raw data all data and daily anomalies the values during the events? It may be helpful to stick to the same wording throughout the paper.
=> Well noted, we will improve the description of Figure 4. The anomalies correspond to NEP values computed on the atmospheric variables after removing the seasonal pattern, see L 209-210: "In order to account for potential seasonality, probabilities of nonexceedance are computed on either the raw daily data or the daily anomalies (substracting the daily climatology from the daily data)". Thus we are able to distinguish the atmospheric variables that are extreme in absolute values (raw data) and those that are extreme for the season (anomalies).

- The Figures 6-8 are very interesting. Their description could be made clearer and more general, to directly make the point why they are shown. With a clear description, the little conclusion (line 236-333) should not be necessary anymore. Right now, it helps understanding the point, but the point should be clear from the beginning of the Figure description.

=> Thank you for this comment. These figures will be moved to the discussion section to give them more weight and the corresponding text will be rewritten.

- Figure 9 could use a clearer description of the "best" variables. Maybe the most discriminant? This choice does however limit comparability between the weather type classes and atmospheric parameters. Why is the colour scale not kept the same? Does it not also say something about which parameters function better during some weather type classes than in others?
=> You're right, the "best" variables are the most discriminant ones; this will be corrected throughout the text. Also the same color scale will be used for all the classes to better hight the couple of variables that are concomitantly abnormal during the events.

Appendix A:
- Line 411: "We keep to alone.." This probably is an old remainder.
=> Sorry, we mean "we keep it alone"

Figures:
- Fig. 1: Maybe something to consider is, that the Figure is difficult to read, when printed in black and white. Please check the Figures following the HESS standards.
=> Well noted, we will check the standards.

- Subplot letters (a, b, c, …) would be handy
=> Well noted

References
Blanc, A.; Blanchet, J. & Creutin, J.-D.
Past evolution of western Europe large-scale circulation and link to precipitation trend in the northern French Alps
*Weather and Climate Dynamics,* **2022***, 3,* 231-250

Dayan, U.; Nissen, K. & Ulbrich, U.
Review Article: Atmospheric conditions inducing extreme precipitation over the eastern and western Mediterranean
*Natural Hazards and Earth System Sciences,* **2015***, 15,* 2525-2544

Grimalt-Gelabert, M.; Alomar-Garau, G. & Martin-Vide, J.
Synoptic Causes of Torrential Rainfall in the Balearic Islands (1941–2010)
*Atmosphere,* **2021***, 12*

Marra, F.; Nikolopoulos, E.; Creutin, J. & Borga, M.
Space–time organization of debris flows-triggering rainfall and its effect on the identification of the rainfall threshold relationship
*Journal of Hydrology,* **2016***, 541,* 246 - 25

Millán, M.; Estrela, M. & Caselles, V.
Torrential precipitations on the Spanish east coast: The role of the Mediterranean sea surface temperature
*Atmospheric Research,* **1995***, 36,* 1-16

Prenner, D.; Hrachowitz, M. & Kaitna, R.
Trigger characteristics of torrential flows from high to low alpine regions in Austria
*Science of The Total Environment,* **2019***, 658,* 958-972